# Nintedanib in Idiopathic Pulmonary Fibrosis: Tolerability and Safety in a Real Life Experience in a Single Centre in Patients also Treated with Oral Anticoagulant Therapy

**DOI:** 10.3390/ph16020307

**Published:** 2023-02-16

**Authors:** Barbara Ruaro, Ilaria Gandin, Riccardo Pozzan, Stefano Tavano, Chiara Bozzi, Michael Hughes, Metka Kodric, Rossella Cifaldi, Selene Lerda, Marco Confalonieri, Elisa Baratella, Paola Confalonieri, Francesco Salton

**Affiliations:** 1Pulmonology Unit, Department of Medical Surgical and Health Sciences, University Hospital of Cattinara, University of Trieste, 34149 Trieste, Italy; 2Biostatistics Unit, Department of Medical Sciences, University of Trieste, 34149 Trieste, Italy; 3Division of Musculoskeletal and Dermatological Sciences, Faculty of Biology, Medicine and Health, The University of Manchester & Salford Royal NHS Foundation Trust, Manchester M6 8HD, UK; 424ORE Business School, Via Monte Rosa, 91, 20149 Milano, Italy; 5Department of Radiology, Cattinara Hospital, University of Trieste, 34127 Trieste, Italy

**Keywords:** idiopathic pulmonary fibrosis (IPF), interstitial pneumonia, interstitial lung disease (ILD), high-resolution computed tomography (HRTC)

## Abstract

Idiopathic pulmonary fibrosis (IPF) is a rare and severe disease with a median survival of ~3 years. Nintedanib (NTD) has been shown to be useful in controlling interstitial lung disease (ILD) in IPF. Here we describe the experience of NTD use in IPF in a real-life setting. Objective. Our objective was to examine the safety profile and efficacy of nintedanib even in subjects treated with anticoagulants. Clinical data of patients with IPF treated with NTD at our center were retrospectively evaluated at baseline and at 6 and 12 months after the introduction of NTD. The following parameters were recorded: IPF clinical features, NTD tolerability, and pulmonary function tests (PFT) (i.e., Forced Vital Capacity (FVC) and carbon monoxide diffusing capacity (DLCO)). In total, 56 IPF patients (34% female and 66% male, mean onset age: 71 ± 11 years, mean age at baseline: 74 ± 9 years) treated with NTD were identified. At enrollment, HRCT showed an UIP pattern in 45 (80%) and a NSIP in 11 (20%) patients. For FVC and FEV1 we found no significant change between baseline and 6 months, but for DLCO we observed a decrease (*p* = 0.012). We identified a significant variation between baseline and 12 months for FEV1 (*p* = 0.039) and for DLCO (*p* = 0.018). No significant variation was observed for FVC. In the cohort, 18 (32%) individuals suspended NTD and 10 (18%) reduced the dosage. Among individuals that suspended the dosage, 14 (78%) had gastrointestinal (GI) collateral effects (i.e., diarrhea being the most common complaint (67%), followed by nausea/vomiting (17%) and weight loss (6%). Bleeding episodes have also not been reported in patients taking anticoagulant therapy. (61%). One patient died within the first 6 months and two subjects died within the first 12 months. In a real-life clinical scenario, NTD may stabilize the FVC values in IPF patients. However, GI side effects are frequent and NTD dose adjustment may be necessary to retain the drug in IPF patients. This study confirms the safety of NTD, even in patients treated with anticoagulant drugs.

## 1. Introduction

Idiopathic pulmonary fibrosis (IPF) is a rare and irreversible chronic interstitial pneumonia characterized by abnormal deposition of extracellular matrix in the lung parenchyma [1,2,3]. Aberrant lung repair leads to repetitive tissue scar formation, alveolar structure abnormalities with significant impairment of alveolar gas exchange, and reduced lung function (forced vital capacity (FVC) and carbon monoxide diffusing capacity (DLCO)) [2,3]. IPF is the most common of its class, being approximately 50–60% of all idiopathic interstitial pneumonia, with an estimated incidence of 3–9 cases per 100,000 individuals in Europe and North America [1,2,3]. IPF patients present a nonspecific symptomatology, which is the fundamental cause of the delay in diagnosis. It causes progressive exertional dyspnea, nonproductive cough and asthenia. These symptoms cause a reduction in daily physical activity and muscle strength; patients suffer a precarious quality of life, which can be responsible of social isolation, depressive and anxiety disorders and increased levels of dependence. The evolution of the disease is often associated with different cardiopulmonary complications. All of these elements ends in a situation that is difficult to manage for both patients and their caregivers [1,2,3,4,5,6]. Another cause for the delay in the diagnosis of IPF is that this is an entity that can be easily confused with other respiratory pathologies requiring multidisciplinary assessment by the pulmonology, radiology and pathological anatomy services, thereby using more healthcare resources [3,4,5,6,7,8,9,10,11]. Although IPF exhibits a varied nature [4], clinical-pathologic and histologic patterns allow the description of so-called habitual interstitial pneumonia (UIP) [5], defined radiologically on high-resolution computed tomography (HRCT) by a bilateral peripheral distribution of fibrosis and most pronounced at the bases of the lungs. The pathogenesis of these patterns is not entirely clear. The environment (cigarette smoke, air pollution, inhalation of wood and metal dust, GERD) plays a significant role in the development of IPF, although without an established relationship to genetics. The single nucleotide polymorphism in the mucin 5B promoter region (MUC5B) is the most relevant genetic risk factor for sporadic and familial IPF, but the mechanisms in lung epithelial cells still need to be better understood. The interaction between variable expression of genetic polymorphisms seems to be the origin of the disease, which is clearly associated also with changes related to cellular aging and exposure to environmental factors, smoking, industrial powders, chronic gastric microaspiration, viral infections and possibly alterations in the lung microbiome [1,3,12,13,14,15,16,17,18,19]. The repetitive exposures aberrantly activate the alveolar epithelial cells of genetically susceptible individuals, promoting epitelial apoptosis, recruitment of mesenchymal cells and increased vascular permeability. Unregulated epithelial/mesenchymal interaction results in the secretion of various profibrotic cytokines, metalloproteinases and procoagulant mediators, which promote uncontrolled migration and proliferation, and differentiation in fibroblasts to myofibroblasts as well as fibrosis in the extracellular matrix [20,21,22,23,24,25,26,27,28]. The main pro-inflammatory cytokines involved in fibrosis are tumour necrosis factor (TNF)-α and interleukin (IL)-1, as well as some fibrous factors such as transforming growth factor (TGF)-β and platelet-derived growth factor (PDGF) [28,29,30,31,32].

Antifibrotic therapies (pirfenidone, nintedanib) can reduce respiratory function decline and improve survival of IPF patients, but they fail to halt disease progression; therefore, prognosis remains poor, with a median survival usually 2–3 years from the time of diagnosis [6,7].

Nintedanib (ex-BIBF 1120) is an intracellular inhibitor of multiple receptor tyrosine kinases (RTKs) and non-receptor tyrosine kinases (nRTKs). It competitively binds to the adenosine triphosphate (ATP) binding pocket of its targets, which include platelet-derived growth factor (PDGFRα and PDGFRβ), fibroblast growth factor receptor (FGFR1, FGFR2, FGFR3), vascular endothelial growth factor (VEGFR1, VEGFR2, VEGFR3), colony-stimulating factor receptor 1 (CSF1R), and FMS-like tyrosine kinase three (FLT3), all of which are critical for fibroblast proliferation, migration, and differentiation [8]. As a result of this broad inhibition, pro-fibrotic and pro-angiogenic processes are reduced and, as a result, fibroblast and myofibroblast number and activity are reduced, as well as ECM (Extracellular Matrix) production, ultimately rebalancing pro-fibrotic and antifibrotic processes [9]. The current indications for the antifibrotic effect of nintedanib are idiopathic pulmonary fibrosis (IPF) from 2014, systemic sclerosis-associated interstitial lung disease (Sc-ILD) from 2019 [5], further confirmed by the SENSCIS and SENSCIS-ON clinical trials [6,7] and progressive pulmonary fibrosis (PPF) from 2020 [10]. In addition, nintedanib is also being used and studied for cancers (lung, ovary, kidney, colorectal, and liver) [11,12,13]; in this context, new modes of administration, such as inhalation [12], are being investigated. The efficacy of nintedanib′s antifibrotic properties first emerged in the phase II TOMORROW study [14] and was later confirmed by the phase III IMPULSIS-1 and IMPULSIS-2 studies [15]. Nintedanib is administered orally at a dosage of 150 mg twice daily or, in cases of poor tolerance, 100 mg twice daily. It reaches maximum plasma concentrations in about 2–4 h depending on whether or not it is taken with food, and reaches steady-state plasma concentrations within a week. The absolute bioavailability of nintedanib 100 mg is 4.7 percent. Biotransformation of nintedanib mainly involves esterase-mediated hydrolysis, followed by glucuronidation, finally yielding glucuronide BIBF 1202. A minor role is played by CYP pathways, particularly CYP 3A4. Caution should be exercised with concomitant administration of nintedanib with CYP3A4 and P-gp inhibitors (e.g., ketoconazole) with inducers (e.g., rifampin), which may result in increased or decreased exposure to nintedanib, respectively. The main route of elimination is fecal/biliary excretion (about 93% of the administered dose), while renal excretion contributes little to the total clearance [1,14,15,16,17,18]; the resulting half-life is 9.5 h. The main adverse effects include, first and foremost, diarrhea (61.5 percent of cases), but also increased liver enzymes (which should be checked before initiating therapy and then monitored), abdominal pain, nausea, vomiting, and weight loss, all of which are generally manageable by reducing the dose (200 mg/day), discontinuing treatment, and applying symptomatic measures (e.g., loperamide) [19,20,21,22,23,24,25,26,27].

The purpose of our retrospective study was to examine the safety profile and efficacy of nintedanib in patients with IPF, including in subjects treated with anticoagulants.

## 2. Results

In this single-center retrospective study, 56 patients with IPF were included: 19 (34%) males, mean age of onset: 71 ± 11 years, mean age at baseline: 74 ± 9 years (Table 1). The prevalence of coexisting GERD was 12 out of 56 patients, with a comparable prevalence of PPI use. Among all comorbidities, smoking was largely prevalent (34 out of 56). Other notable comorbidities were COPD (5.5%) and cancer (15%). The most frequent clinical symptoms were cough (80% of patients) and dyspnea (50%). Concomitant medications were considered: steroid use was the most frequent (40%), especially at a high steroid dosage of >15 mg (21%). A minority of patients were on immunosuppressants, specifically two patients on hydroxychloroquine, three on mycophenolate mofetil, one on azathioprine, and one on rituximab.

All 56 patients started NTD therapy at baseline, of whom 25 (45%) completed the 12-month follow-up at full dose, 10 (18%) reduced the dose and 18 (32%) discontinued the drug. In addition, two patients (3.6%) died within 12 months and one (1.8%) died within 6 months. The main reason for treatment reduction was gastrointestinal intolerance, particularly diarrhea (six patients), but also nausea (one patient), and abdominal pain (one patient). Second, liver disruption resulted in four NTD dose reductions. Regarding discontinuation, 14 of 18 discontinued treatment due to gastrointestinal intolerance, particularly diarrhea (12 patients) but also nausea/vomiting (3 patients). One patient reported weight loss and consequently discontinued the drug. No hemorrhagic events were reported in our population even in patients taking anticoagulants.

Pulmonary function test (PFT) data were included, specifically Forced Vital Capacity (FVC), Forced Expiratory Volume in 1 s (FEV1), and Diffusion Lung CO (DLCO) were recorded at baseline, 6 months, and 12 months (Table 1). FVC measurements at baseline and 6 months were available for 31 patients, with no statistically significant differences (*p*-value 0.6, Figure 1). FEV1 measurements at baseline and 6 months were available for 29 subjects, with no statistically significant differences (*p*-value > 0.9). In contrast, DLCO measurements, available at baseline and at 6 months for 25 patients, revealed a statistically significant difference (*p*-value 0.012).

The same evaluation was made for differences between measurements at baseline and 12 months, which for FVC was available for 22 patients, for FEV1was available for 22 patients, and for DLCO was available for 20 patients (Figure 2 and Figure 3). In this case, a statistically significant difference was detected for both FEV1 and DLCO (*p*-value, respectively, 0.039 and 0.018, Figure 2), while difference in FVC between baseline and 12 months expressed no statistical significance (*p*-value 0.2). In Figure 3, it is possible to observe the trend over the three time-point for subjects who had all the information available. The analysis restricted to male individuals gave similar results. Both in the 6-month and 12-month change, we only observed a significant decrease for DLCO (*p* = 0.018 and *p* = 0.020, respectively).

For FVC, six patients (19%) were identified as non-responders at 6 months and three (14%) as non-responders at 12 months (Table 2). No association was found with smoking status, steroid use, anti-antibody use, or PPI. For DLCO, 13 (52%) were identified as non-responders at 6 months and 5 (25%) at 12 months. Similarly, in this case no association was found with smoking status, steroid use, anti-antibody use, and PPI.

Considering individuals that took NTD for at least 4 months, life status information was available for 50 of them, with median follow-up of 17 months. The overall survival was 80.1% (95% CI [68.7, 80.1]) at one year, 60.8% (95% CI [47.0, 78.6]) at two years, and 38.4% (95% CI [24.8, 59.4]) at three years (Figure 4).

## 3. Discussion

This retrospective single-center study aims to evaluate the role of NTD in the treatment of 56 patients with IPF. Our study confirms the efficacy of NTD in slowing disease progression in terms of stabilizing lung function without, as already known, allowing a marked improvement in fibrotic damage. Patients with IPF may have a variable clinical course also influenced by comorbidities, smoking habits first and foremost, which affected 61% of the enrolled patients and by any NTD-associated therapies; however, the existence of a phenotype associated with a rapid decline in FVC leading rapidly to death, which occurred within a year in 5% of the cases in our cluster, should also be kept in mind [22,23,24,25]. In the studies performed in the 1990s and the early 2000s, before the era of the new American Thoracic Society (ATS)/European Respiratory Society (ERS) classification and the introduction of antifibrotic medication, the median survival of IPF patients was 2–3 years [26,27,28]. Several recent studies demonstrated that IPF mortality is increasing in Europe [26,27,28]. The Finnish IPF registry enrolled 453 patients; at the time of diagnosis, the mean age of these patients was 73.0 years and the age varied from 44 to 91 years [26]. The 5-year survival rate was 45%. The independent predictors of survival were age and lung function at diagnosis. During 2011–2017, only a quarter of the patients had received antifibrotic medication, and the transplant rate was very low. Patients who received ≥6 months of treatment had better survival compared with those who did not receive treatment [26]. In Italy, the last official data before the COVID-19 pandemic show that in healthy people with similar age of IPF patients, a 70-year-old male today has about a 33% chance of dying by age 80, a 50% chance by age 84, and a 20% chance of reaching 90. A 70-year-old woman has a more than 80% chance of making it to age 80, a 50% chance of making it to 87 and a nearly 20% chance of making it to 93 [29].

On the SENSCIS line, about half of the patients examined were on NTD monotherapy; the remainder were on concomitant treatment with OCS and only a minority were on immunosuppressants. In our study, only 13% could be classified into this subgroup. No major adverse events were reported in this group of patients treated with this combination. This finding highlights that NTD can currently be considered a single reference therapy in the treatment of IPF. We found no association between the non-responders status and smoking status, anti-antibody, PPI, and both steroid groups, at 6 and 12 months, respectively, for all spirometry parameters. Our study considered patients without distinction of age or exclusion criteria. The number of patients enrolled was lower than that of another previous study, which examined two groups of patients divided according to age above or below 80 years, with a single collection of functional data at 12 months [24]. From the point of view of pulmonary function, the most significant data in our study concerned FEV1 values at 6 months and DLCO values at 6 and 12 months, measured for a sample of about 20 follow-up patients. As is well known, DLCO is a typically altered parameter in IPF and is closely related to the development of other comorbidities, such as pulmonary hypertension. The stabilization of diffusing capacity values after the initiation of NTD therapy confirms what is expected in terms of treatment efficacy. Moreover, most of our patients had UIP pathology at baseline at HRTC (80%), which could explain the essentially stable FVC values during follow-up. In particular, the systematic review and network meta-analysis conducted by Pitre et al. [23], highlights not only the potential of NTD in reducing FVC decline (2.92%; 1.51 to 4.14), but also its efficacy in reducing mortality (RR 0.69; 0.44 to 1.1). One noteworthy element concerns the safety of NTD therapy in enrolled patients. As in the SENSCIS study, gastrointestinal adverse effects (AE) were the main problem associated with NTD. The percentage of patients who reduced drug dosage was lower (18% vs. 48% over 12 months), while only 32% of subjects discontinued treatment. Given the expected results in terms of lung function, it was our specific intention to reduce the dosage as soon as possible (due to difficult gastrointestinal tolerance, rather than in case of increased liver function test), rather than to recommend permanent discontinuation of treatment, so as not to lose the patient during the study. The safety of this drug regimen confirmed by our study is in line with evidence from the latest work on the subject. In fact, analysis of the Latin American Registry of IPF [24] showed that NTD therapy leads to AE in a third of patients, with a non-statistically significant difference in terms of the proportion of patients with AE compared with pirfenidone. Furthermore, this study confirms the safety of NTD even in patients treated with anticoagulant drugs. The main limitation of our study is a limited number of enrolled patients, as well as the even smaller number of repeated measurements. Despite this, it was possible to obtain evidence on the efficacy of NTD in terms of progression of the disease from a pulmonary function point of view as expected. Finally, it was not possible to assess whether stabilization of lung function was associated with better quality of life because of the absence of a systematic evaluation of this parameter.

In the future, the increase in the numbers of patients enrolled, and above all the availability of PTF data—in particular about the relation to the variation of FVC over time—would make it possible to analyze the trend of lung function together with main clinical factors (e.g., using a multivariate model). This could surely help to define more precisely the effective therapeutic efficacy of NTD.

## 4. Materials and Methods

### 4.1. Study Population

Patients classified as IPF according to the 2022 ATS/ERS/JRS/ALAT Clinical Practice Guideline and who started treatment with nintedanib (NTD) were identified in our center and were retrospectively reviewed.

### 4.2. Data Collection

Clinical and laboratory data were collected retrospectively at baseline and at 6 and 12 months after the introduction of NTD. The following clinical characteristics were collected: sex, age, duration of illness, and clinical symptoms (fever, cough, dyspnea, arthralgia/arthritis). In addition, comorbidities including hypertension, coronary artery disease, diabetes mellitus, chronic lung disease, and rheumatic diseases were recorded. All adverse events were recorded. Ethical approval and informed patient consent were obtained from our center. Laboratory results included blood cell counts (neutrophils, lymphocytes, hemoglobin, platelets); biochemical parameters (aspartate aminotransferase, AST; lactic dehydrogenase, LDH; creatine kinase, CK); erythrocyte sedimentation rate (ESR); and C-reactive protein. Antinuclear antibodies (ANA) and extractable nuclear antigens (ENA) were also collected (Table 3).

### 4.3. Assessment of Imaging Involvement

The lung disease presented mainly as interstitial lung disease (ILD) with a variety of radiological pictures, which included usual interstitial pneumonia (UIP), nonspecific interstitial pneumonia (NSIP), and organizational pneumonia (OP) [4]. The evaluation of ILD in our hospital was performed by radiologists with more than 10 years of experience in chest imaging, relying mainly on clinical manifestations and respective abnormalities suggestive of ILD in high-resolution computed tomography (HRCT). During the data review, any indeterminate ILD was excluded from the analysis.

### 4.4. Assessment of Pulmonary Involvement

Pulmonary function tests (PFT) were performed among patients with ILD, and the following indexes were recorded, including forced vital capacity (FVC), the median forced expiratory volume in 1 s (FEV1), and diffusing capacity of the lung for carbon monoxide (DLCO). FVC, FEV1, and DLCO were described as a percentage of the predicted values for the patient’s age, sex, and height [21]. Abnormalities of PFTs were defined as predicted values of forced vital capacity (FVC) < 80% and diffusing capacity of the lung for carbon monoxide (DLCO) < 70% [22].

### 4.5. Statistical Analysis

Continuous variables were summarized using mean (standard deviation) or median (interquartile range) as appropriate, while categorical variables were reported as percentages. The normality of continuous variables was tested with the Shapiro–Wilk test. As for spirometry measurements, values were compared at different time points using the paired *t*-test or the non-parametric Wilcoxon signed-rank test when assumptions for the parametric test were not satisfied. When comparing measurements at baseline with those at 6 months, we considered only those individuals who had been taking NTDs for at least 4 months. When comparing measurements at baseline and at 2 months, we considered only individuals who had been taking NTD for at least 9 months. The comparison was carried out in the restricted sample of male subjects. For FVC, we considered a non-responder subject after 6 months of NTD if FVC decreased >5% and if differences were observed during the follow-up regarding blood test evaluations.

We considered a non-responder subject after 10 months of NTD if FVC decreased >10%. Similarly, for DLCO, we considered a non-responder subject after 6 months of NTD if DLCO decreased >6% and we considered a non-responder subject after 10 months of NTD if DLCO decreased >10% We performed a survival analysis considering the onset of NTD as a baseline and after 1 year of observation. The endpoint was death from any cause. Survival function was estimated by the Kaplan–Meier method. In all the analysis, the statistical significance was set at 0.05. Analyses were performed using R software version 4.2.1.

## 5. Conclusions

Our real-life study, despite the small number of patients enrolled, confirms the efficacy and safety of NTD. This therapy aims to stabilize lung function in terms of FVC values, although it is unable to achieve a marked improvement in fibrosis. Our relatively small study population confirms the effect of NTD in stabilizing FVC; furthermore, it does confirm its major role in slowing DLCO decline, which is another pivotal goal of this therapy. Despite the retrospective design and its inherent limitations, this study is of importance because real-life data are expected to increase the knowledge about NTB effects.

Indeed, in this observational study, the results report some meaningful effects with stabilization of FVC in UIP patients and a safety profile that sounds better than the one reported in SENSCIS trial. Furthermore, our data supported good safety, even in patients on anticoagulants.

In the future, increasing the number of patients enrolled and the data collected may better clarify the actual therapeutic efficacy of NTD.

## Figures and Tables

**Figure 1 pharmaceuticals-16-00307-f001:**
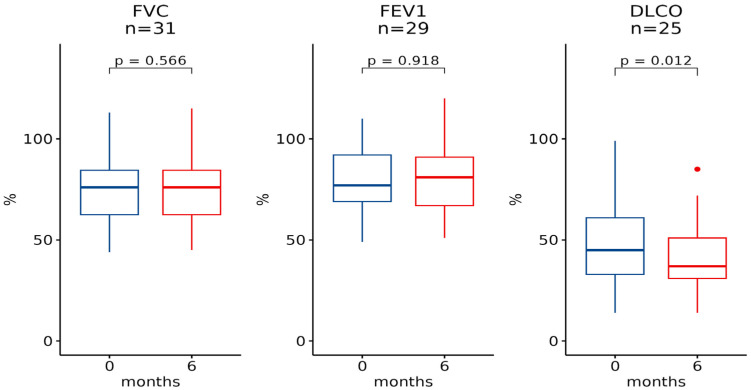
The 6-month trend for spirometry measurements. Box-plots representing levels of FVC, FEV1 and DLCO at baseline and after 6 months.

**Figure 2 pharmaceuticals-16-00307-f002:**
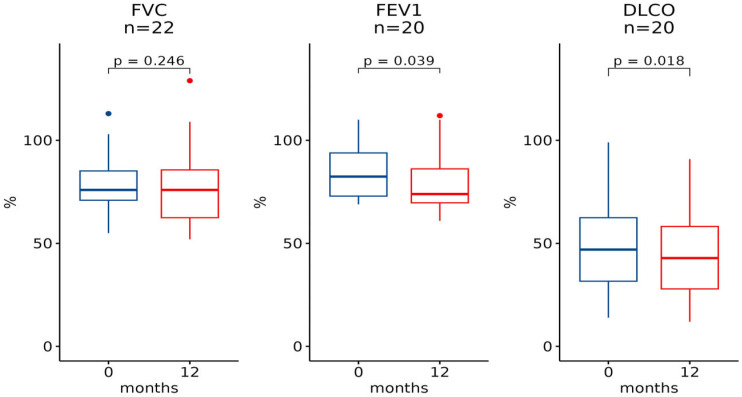
The 12-month trend for spirometry measurements. Box-plots representing levels of FVC, FEV1 and DLCO at baseline and after 12 months.

**Figure 3 pharmaceuticals-16-00307-f003:**
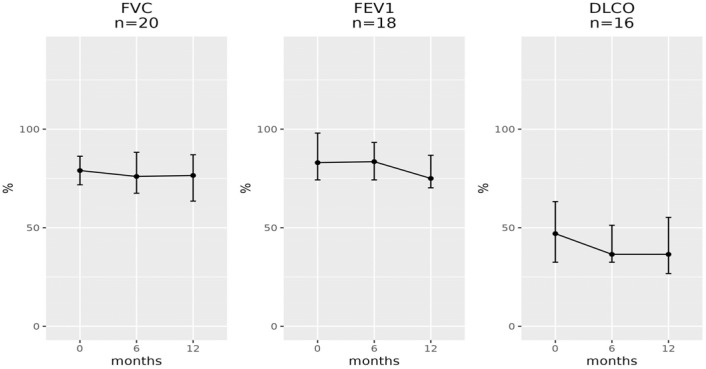
Trend for spirometry measurements between 0 and 12 months. Dots represent the median value, bars indicate the first (Q1) and the third (Q3) quartiles.

**Figure 4 pharmaceuticals-16-00307-f004:**
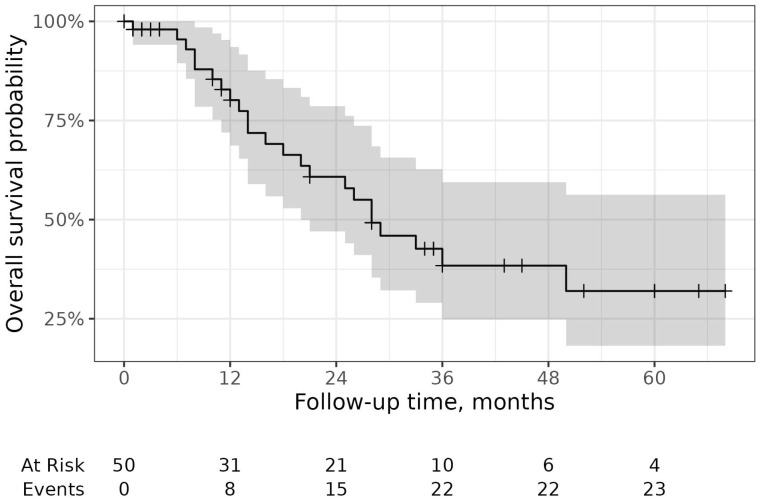
Overall survival. On the top, Kaplan–Meier curve for overall survival. On the bottom, number of subject at risk and number of events for each time point.

**Table 1 pharmaceuticals-16-00307-t001:** Comparison of spirometric measurements after 6 months and after 12 months.

Baseline [Median (IQR)]	6 Months [Median (IQR)]	n	*p*-Value	Baseline [Median (IQR)]	12 Months [Median (IQR)]	n	*p*-Value
FVC							
76.0 (22.0)	76.0 (22.0)	31	0.6	76.0 (14.2)	76.0 (23.2)	22	0.2
FEV1							
77.0 (23.0)	81.0 (24.0)	29	>0.9	82.5 (21.0)	74.0 (16.5)	20	0.039
DLCO							
45.0 (28.0)	37.0 (20.0)	25	0.012	47.0 (30.8)	43.0 (30.2)	20	0.018

**Table 2 pharmaceuticals-16-00307-t002:** Steroid use in responders and non-responders for FVC and DLCO.

FVC					
6 months			12 months		
Responder (n = 25)	Non-responder (n = 6)	*p*-value	Responder (n = 19)	Non-responder (n = 3)	*p*-value
10 (40%)	2 (33%)	>0.9	7 (39%)	2 (67%)	0.6
DLCO					
6 months			12 months		
Responder (n = 12)	Non-responder (n = 13)	*p*-value	Responder (n = 15)	Non-responder (n = 5)	*p*-value
4 (33%)	6 (46%)	0.7	4 (29%)	4 (80%)	0.1

**Table 3 pharmaceuticals-16-00307-t003:** Clinical and demographic characteristics of our cohort of IPF patients.

Clinical and demographic characteristics	56 patients
Females, n (%)	19 (34%)
Mean onset age (years), mean ± SD	71 ± 11 years
Mean age at baseline (years), mean ± SD	74 ± 9 years
Smokers (active or former)	34 (62%)
Autoantibodies	
Antinuclear antibodies (ANA)	7 (12%)
Anti-Ro52, n (%)	3 (5%)
Organ involvement	
COPD (n, %)	3 (5%)
Cancer (n, %)	8 (15%)
Hypertension (n, %)	36 (64%)
Coronary heart disease (n, %)	11 (19%)
Diabetes mellitus (n, %)	22 (39%)
Rheumatic diseases (n, %)	1 (2%)
Previous episodes of venous thromboembolism (n, %)	11 (19%)
Cardiac comorbidities (n, %)	31 (55%)
Concurrent Therapies	
Mycophenolate mofetil, n (%)	3 (5%)
Azathioprine, n (%)	1 (2%)
Hydroxychloroquine, n (%)	2 (4%)
Rituximab, n (%)	1 (2%)
Steroid high dose (i.e., >15 mg/die), n (%)	12 (21%)
Steroid low dose (i.e., <15 mg/die), n (%)	8 (15%)
Antihypertensive drugs, n (%)	36 (64%)
Pump inhibitor drugs, n (%)	38 (59%)
Anticoagulant drugs, n (%)	14 (25%)
Warfarin, n (%)	11 (20%)
Apixaban and Rivaroxaban	3 (5%)
Baseline HRCT	56 patients
NSIP, n (%)	11 (20%)
UIP, n (%)	45 (80%)

Legend. N = number; SD = standard deviation; Ab = antibodies; GI = gastro-intestinal; NSIP = non-specific interstitial pneumonia; UIP = usual interstitial pneumonia.

## Data Availability

Deidentified participant data will be made available upon motivated request to the Corresponding Author.

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
