# Peer review of "Nintedanib in Idiopathic Pulmonary Fibrosis: Tolerability and Safety in a Real Life Experience in a Single Centre in Patients also Treated with Oral Anticoagulant Therapy"

_pharmaceuticals, 2023, doi:10.3390/ph16020307_

Round 1

Reviewer 1 Report

Ruaro et al. reported a retrospective evaluation of Nintedanib in idiopathic pulmonary fibrosis using a single agent in 56 IPF patients. Currently, all Nintedanib data are well known to clinicians, and this manuscript does not provide new information to the reader. Curiously, the authors only administered a single medication in their clinic, with no controls. As an assessment assay, the conclusions are weak. 

Author Response

Thank you for your letter and for the reviewer's comments on our manuscript titled “Nintedanib in Idiopathic Pulmonary Fibrosis: Tolerability and Safety in a Real-Life Single Center Experience in Patients Also Treating Oral Anticoagulant Therapy.” These comments are all valuable and very helpful in the review and improvement of our article as well as important guiding significance for our research. We have carefully reviewed the comments and made a correction, which we hope will meet your approval. The main corrections in the document and the answers to the reviewer's questions and comments are as follows.

Ruaro et al. reported a retrospective evaluation of Nintedanib in idiopathic pulmonary fibrosis using a single agent in 56 IPF patients. Currently, all Nintedanib data are well known to clinicians, and this manuscript does not provide new information to the reader. Curiously, the authors only administered a single medication in their clinic, with no controls. As an assessment assay, the conclusions are weak. 

R: As suggested, we improve several parts of our manuscript and we add numerous figures and table for underlining that our study confirm that NTD may stabilize the FVC values in IPF patients also in a real-life clinical situations. Furthermore, this observation confirms the safety of NTD even in patients treated with anticoagulant drugs. In addition, we underline that NTD dose adjustment may be useful in case of GI side.

“1. INTRODUCTION. Idiopathic pulmonary fibrosis (IPF) is a rare and nonreversible chronic interstitial pneumonia characterized by abnormal deposition of extracellular matrix in the lung parenchyma [1-3]. Aberrant lung repair leads to repetitive tissue scar formation, alveolar structure abnormalities with significant impairment of alveolar gas exchange, and reduced lung function [forced vital capacity (FVC) and carbon monoxide diffusing capacity (DLCO)] [2,3]. IPF is the most common of its class (approximately 50-60% of all idiopathic interstitial pneumonia, with an estimated incidence of 3-9 cases per 100,000 individuals in Europe and North America [1-3]. It causes progressive exertional dyspnea and nonproductive cough, with rapid deterioration in the quality of life of patients with IPF. Although IPF exhibits a varied nature [4], clinical-pathologic and histologic patterns allow the description of so-called habitual interstitial pneumonia (UIP) [5], defined radiologically on high-resolution computed tomography (HRCT) by a bilateral peripheral distribution of fibrosis and most pronounced at the bases of the lungs. The pathogenesis of these patterns is not entirely clear. The environment (cigarette smoke, air pollution, inhalation of wood and metal dust, GERD) plays a significant role in the development of IPF, although without an established relationship to genetics. The single nucleotide polymorphism in the mucin 5B promoter region (MUC5B) is the most relevant genetic risk factor for sporadic and familial IPF, but the mechanisms in lung epithelial cells still need to be better understood. Antifibrotic therapies (pirfenidone, nintedanib) can reduce respiratory functional decline and improve survival of IPF patients, but they fail to halt disease progression; therefore, prognosis remains poor, with median survival usually 2-3 years from the time of diagnosis [6,7]. Nintedanib (ex-BIBF 1120) is an intracellular inhibitor of multiple receptor tyrosine kinases (RTKs) and nonreceptor tyrosine kinases (nRTKs). It competitively binds to the adenosine triphosphate (ATP) binding pocket of its targets, which include plaque-derived growth factor (PDGFRα and PDGFRβ), fibroblast growth factor receptor (FGFR1, FGFR2, FGFR3), vascular endothelial growth factor (VEGFR1, VEGFR2, VEGFR3), colony-stimulating factor receptor 1 (CSF1R), and Fms-like tyrosine kinase3 (FLT3), all of which are critical for fibroblast proliferation, migration, and differentiation [8]. As a result of this broad inhibition, pro-fibrotic and pro-angiogenic processes are reduced and, as a result, fibroblast and myofibroblast number and activity are reduced, as well as ECM (ExtraCellular Matrix) production, ultimately rebalancing pro-fibrotic and antifibrotic processes [9]. The current indications for the antifibrotic effect of nintedanib are idiopathic pulmonary fibrosis (IPF), from 2014, systemic sclerosis-associated interstitial lung disease (Sc-ILD), from 2019 [5], further confirmed by the SENSCIS and SENSCIS-ON clinical trials [6,7] and progressive pulmonary fibrosis (PPF), from 2020 [10]. In addition, nintedanib is also being used and studied for cancers (lung, ovary, kidney, colorectum, and liver) [11-13]; in this context, new modes of administration, such as inhalation [12], are being investigated. The efficacy of nintedanib's antifibrotic properties first emerged in phase II TOMORROW study [14] and was later confirmed by the phase III IMPULSIS-1 and IMPULSIS-2 studies [15]. Nintedanib is administered orally at a dosage of 150 mg twice daily or, in cases of poor tolerance, 100 mg twice daily. It reaches maximum plasma concentrations in about 2-4 hours (depending on whether or not it is taken with food) and steady-state plasma concentrations within a week. The absolute bioavailability of nintedanib 100 mg is 4.7 percent. Biotransformation of nintedanib mainly involves esterase-mediated hydrolysis, followed by glucuronidation, finally yielding the glucuronide BIBF 1202. A minor role is played by CYP pathways, particularly CYP 3A4. Caution should be exercised with concomitant administration of nintedanib with CYP3A4 and P-gp inhibitors (e.g., ketoconazole) and inducers (e.g., rifampin), which may result in increased or decreased exposure to nintedanib, respectively. The main route of elimination is fecal/biliary excretion (about 93% of the administered dose), while renal excretion contributes little to the total clearance [1,14-16]; the resulting half-life is 9.5 hours. The main adverse effects include first and foremost diarrhea (61.5 percent of cases), but also increased liver enzymes (which should be checked before initiating therapy and then monitored), abdominal pain, nausea, vomiting, and weight loss, all of which are generally manageable by reducing the dose (200 mg/day), discontinuing treatment, and applying symptomatic measures (e.g., loperamide) [17-22]. The purpose of our retrospective study was to examine the safety profile and efficacy of nintedanib in patients with IPF, including in subjects treated with anticoagulants”.  “2.5 Statistical analysis. Continuous variables were summarized using mean (standard deviation) or median (interquartile range) as appropriate, while categorical variables were reported as percentages. The normality of continuous variables was tested with the Shapiro-Wilk test. As for spirometry measurements, values were compared at different time points using the paired t-test or the nonparametric Wilcoxon signed-rank test, when assumptions for the parametric test were not satisfied. When comparing measurements at baseline with those at 6 months, we considered only those individuals who had been taking NTDs for at least 4 months. When comparing measurements at baseline and at 2 months, we considered only individuals who had been taking NTD for at least 9 months. The comparison was carried in the restricted sample of male subjects. For FVC, we considered a non-responder subject after 6 months of NTD if FVC decreased > 5% and considered a differences were observed during the follow-up regarding blood test evaluations. Concomitant non-responder subject after 10 months of NTD if FVC decreased > 10%. Similarly, for DLCO we considered a non-responder subject after 6 months of NTD if DLCO decreased > 6% and we considered a non-responder subject after 10 months of NTD if DLCO decreased > 10% We performed a survival analysis considering the onset of NTD as a baseline and after 1 year of observation. The endpoint was death from any cause. Survival function was estimated by the Kaplan-Meier method. In all the analysis, the statistical significance was set at .05. Analyses were performed using R software version 4.2.1”.

“The same evaluation was made for differences between measurements at baseline and 12 months, which for FVC were available for 22 patients, for FEV1 for 22 patients and for DLCO for 20 patients (Figure 2 and 3). In this case, a statistically significant difference was detected for both FEV1 and DLCO (p-value respectively 0.039 and 0.018, Figure 2), while difference in FVC between baseline and 12 months expressed no statistical significance (p-value 0.2). In Figure 3 it is possible to observe the trend over the three time-point for subjects who had all the information available. The analysis restricted to male individuals gave similar results. Both in the 6 month- and 12 month-change, we only observed a significant decrease for DLCO (p=0.018 and p=0.020, respectively). For FVC, 6 (19%) were identified as non-responders at 6 months and 3 (14%) as non-responders at 12 months (Table 3). No association was found with smoking status, steroid, anti-antibody, PPI. For DLCO 13 (52%) were identified as non-responders at 6 months and 5 (25%) at 12 months. Equal in this case no association was found with smoking status, steroid, anti-antibody, PPI.

FVC

6 months

12 months

Responder (n=25)

Non-responder (n=6)

p-value

Responder (n=19)

Non-responder (n=3)

p-value

10 (40%)

2 (33%)

>0.9

7 (39%)

2 (67%)

0.6

DLCO

‍6 months

12 months

Responder (n=12)

Non-responder (n=13)

p-value

Responder (n=15)

Non-responder (n=5)

p-value

‍4 (33%)

6 (46%)

0.7

4 (29%)

4 (80%)

0.1

Table 3. Steroid in responders and non-responders for FVC and DLCO.

Figure 2. The 12-months trend for spirometry measurements. Box-plots representing levels of FVC, FEV1 and DLCO at baseline and after 12 months

Figure 3. Trend for spirometry measurements between 0 and 12 months. Dots represent the median value, bars indicate the first (Q1) and the third (Q3) quartiles.

“4. DISCUSSION. This retrospective single-center study aims to evaluate the role of NTD in the treatment of 56 patients with IPF. Our study confirms the efficacy of NTD in slowing disease progression, in terms of stabilizing lung function, without, however, as already known, allowing a marked improvement in fibrotic damage. Patients with IPF may have a variable clinical course, also influenced by comorbidities, smoking habit first and foremost, which affected 61% of the enrolled patients and by any NTD-associated therapies; however, the existence of a phenotype associated with a rapid decline in FVC leading rapidly to death, which occurred within a year in 5% of the cases in our cluster, should also be kept in mind [22-25]. In the studies realized from the 1990s and the early 2000s, before the era of the new American Thoracic Society (ATS)/European Respiratory Society (ERS) classification and the introduction of antifibrotic medication, the median survival of IPF patients was 2–3 years [26-28]. Several recent studies demonstrated that IPF mortality is increasing in Europe [26-28]. The FinnishIPF registry enrolled 453 patients; at time of diagnosis, the mean age of these patients was 73.0 years and the age varied from 44 to 91 years [26]. The 5-year survival rate was 45%. The independent predictors of survival were age and lung function at diagnosis. During 2011–2017, only a quarter of the patients had received antifibrotic medicines and the transplantation rate was very low. Patients who received ≥6 months of treatment had better survival compared with those who did not receive treatment [26]. In Italy, the last official data before COVID-19 pandemic, show in healthy people with similar age of IPF patients, that a 70-year-old male today has about a third chance of dying by age 80, a 50% chance by age 84, and a 20% chance of reaching 90. A woman 70 years old has more than 80% chance of making it to age 80, 50% of making it to 87 and nearly 20% of making it to 93 [29]. On the SENSCIS line, about half of the patients examined were on NTD monotherapy; the remainder were on concomitant treatment with OCS and only a minority were on immunosuppressants. In our study, only 13% could be classified into this subgroup. No major adverse events were reported in this group of patients treated with this combination. This finding highlights that NTD can currently be considered a single reference therapy in the treatment of IPF. We found no association between the non-responders status and smoking status, anti-antibody, PPI and both steroid groups, at 6 and a 12 months respectively for all spirometry parameters. Our study considered patients without distinction of age or exclusion criteria. The number of patients enrolled was lower than that of another previous study, which examined two groups of patients divided according to age above or below 80 years, with a single collection of functional data at 12 months [24]. From the point of view of pulmonary function, the most significant data in our study concerned FEV1 values at 6 months and DLCO values at 6 and 12 months, measured for a sample of about 20 follow-up patients. As is well known, DLCO is a typically altered parameter in IPF and is closely related to the development of other comorbidities, such as pulmonary hypertension. The finding of stabilization of diffusing capacity values after the initiation of NTD therapy confirms what is expected in terms of treatment efficacy. Moreover, most of our patients had UIP pathology at baseline at HRTC (80%), which could explain the essentially stable FVC values during follow-up. In particular, the systematic review and network meta-analysis conducted by Pitre al.[23], highlights not only the potential of NTD in reducing FVC decline (2.92%; 1.51 to 4.14), but also its efficacy in reducing mortality (RR 0.69; 0.44 to 1.1). One certainly noteworthy element concerns the safety of NTD therapy in enrolled patients. As in the SENSCIS study, gastrointestinal adverse effects (AE) were the main problem associated with NTD. The percentage of patients who reduced drug dosage was lower (18% vs. 48% over 12 months), while only 32% of subjects discontinued treatment. Given the expected results in terms of lung function, it was our specific intention to reduce the dosage as soon as possible (due to difficult gastrointestinal tolerance, rather than in case of increased liver function test), rather than to recommend permanent discontinuation of treatment, so as not to lose the patient during the study. The safety of this drug regimen confirmed by our study is in line with evidence from the latest work on the subject. In fact, analysis of the Latin American Registry of IPF [24] showed that NTD therapy leads to AE in 1/3 of patients, with a non-statistically significant difference in terms of the proportion of patients with AE compared with pirfenidone. Furthermore, this study confirms the safety of NTD even in patients treated with anticoagulant drugs. The main limitation of our study is clearly a limited number of enrolled patients, as well as the even smaller number of repeated measurements. Despite this, it was possible to obtain evidence on the efficacy of NTD in terms of progression of the disease from a pulmonary function point of view as expected. Finally, it was not possible to assess whether stabilization of lung function was associated with better quality of life because of the absence of a systematic evaluation of this parameter. In the future, the increase in the numbers of patients enrolled, but above all the availability of PTF data, in particular in relation to the variation of FVC over time, it would possible to analyze the trend of lung function together with main clinical factors (eg. using a multivariate model) and this could surely help to define more precisely the effective therapeutic efficacy of NTD”.

Reviewer 2 Report

The manuscript by Ruaro et al. investigates the safety profile and efficacy of nintedanib (NTD) in 56 patients with Idiopathic Pulmonary Fibrosis receiving concomitant oral anticoagulation therapy. Forced vital capacity (FVC), median forced expiratory volume in 1 second (FEV1), and diffusing capacity of the lung for carbon monoxide (DLco) were recorded at baseline, 6 months, and 12 months. IPF clinical features and NTD tolerance were also documented. The results suggested that NTD may stabilize FVC and slow DLCO decline. This is an interesting study that confirms the efficacy and safety of NTD treatment, however, some parts do require further details.

Major point:

1.       Figure 1, and 2: what method was used in statistical analysis? what is the number of replicates?

2.       The authors compared baseline and 6-month data, also compared baseline and 12-month data, but did not compare 6 and 12-month data. It would be useful to add a line graph to show baseline, 6-month, and 12-month data.

3.       Consider concomitant medications: steroids were used most frequently (40%). The authors stated that no association with steroids was found. It would be useful to present this data.

4.       Survival survey: 56 IPF patients (average baseline age: 74±9 years old) participated in the survival evaluation. What is the survival rate of healthy people with an average age of 74 at the same follow-up time? Reference support is required if available.

Author Response

Thank you for your letter and for the reviewer's comments on our manuscript titled “Nintedanib in Idiopathic Pulmonary Fibrosis: Tolerability and Safety in a Real-Life Single Center Experience in Patients Also Treating Oral Anticoagulant Therapy.” These comments are all valuable and very helpful in the review and improvement of our article as well as important guiding significance for our research. We have carefully reviewed the comments and made a correction, which we hope will meet your approval. The main corrections in the document and the answers to the reviewer's questions and comments are as follows.

The manuscript by Ruaro et al. investigates the safety profile and efficacy of nintedanib (NTD) in 56 patients with Idiopathic Pulmonary Fibrosis receiving concomitant oral anticoagulation therapy. Forced vital capacity (FVC), median forced expiratory volume in 1 second (FEV1), and diffusing capacity of the lung for carbon monoxide (DLco) were recorded at baseline, 6 months, and 12 months. IPF clinical features and NTD tolerance were also documented. The results suggested that NTD may stabilize FVC and slow DLCO decline. This is an interesting study that confirms the efficacy and safety of NTD treatment, however, some parts do require further details.

Major point:

  1. Figure 1, and 2: what method was used in statistical analysis? what is the number of replicates?

R: We thank the reviewer for all comments. Spirometry measurements were compared at two time points using statistical tests for paired data (t-test or the nonparametric Wilcoxon signed-rank test when assumption for the parametric test were not satisfied). In agreement with reviewer’s observation this information is now included in the caption of the figures, as well as the number of replicates.

  1. The authors compared baseline and 6-month data, also compared baseline and 12-month data, but did not compare 6 and 12-month data. It would be useful to add a line graph to show baseline, 6-month, and 12-month data.

R: As suggested by the reviewer, we included a line graph showing  baseline, 6-month, and 12-month values.

However, it should be noted that the number of subjects presenting all the three measurements is quite low (n=20 for VDC, n=18 for FEV1, n=16 for DLCO).

Figure 3. Trend for spirometry measurements between 0 and 12 months. Dots represent the median value, bars indicate the first (Q1) and the third (Q3) quartiles.

  1. Consider concomitant medications: steroids were used most frequently (40%). The authors stated that no association with steroids was found. It would be useful to present this data.

R: In agreement with the reviewer’s comments, a table (Table 3) reporting use of steroid in responders and non-responders has been added to the main text.

FVC

6 months

12 months

Responder (n=25)

Non-responder (n=6)

p-value

Responder (n=19)

Non-responder (n=3)

p-value

10 (40%)

2 (33%)

>0.9

7 (39%)

2 (67%)

0.6

DLCO

‍6 months

12 months

Responder (n=12)

Non-responder (n=13)

p-value

Responder (n=15)

Non-responder (n=5)

p-value

‍4 (33%)

6 (46%)

0.7

4 (29%)

4 (80%)

0.1

Table 3. Steroid in responders and non-responders for FVC and DLCO.

  1. Survival survey: 56 IPF patients (average baseline age: 74±9 years old) participated in the survival evaluation. What is the survival rate of healthy people with an average age of 74 at the same follow-up time? Reference support is required if available.

R:    We thank the reviewer for this interesting observation. In agreement with the reviewer’s comments, we add this paragraph in the manuscript and some references in the manuscript as follows: “Patients with IPF may have a variable clinical course, also influenced by comorbidities, smoking habit first and foremost, which affected 61% of the enrolled patients and by any NTD-associated therapies; however, the existence of a phenotype associated with a rapid decline in FVC leading rapidly to death, which occurred within a year in 5% of the cases in our cluster, should also be kept in mind [22-25]. In the studies realized from the 1990s and the early 2000s, before the era of the new American Thoracic Society (ATS)/European Respiratory Society (ERS) classification and the introduction of antifibrotic medication, the median survival of IPF patients was 2–3 years [26-28]. Several recent studies demonstrated that IPF mortality is increasing in Europe [26-28]. The FinnishIPF registry enrolled 453 patients; at time of diagnosis, the mean age of these patients was 73.0 years and the age varied from 44 to 91 years [26]. The 5-year survival rate was 45%. The independent predictors of survival were age and lung function at diagnosis. During 2011–2017, only a quarter of the patients had received antifibrotic medicines and the transplantation rate was very low. Patients who received ≥6 months of treatment had better survival compared with those who did not receive treatment [26]. In Italy, the last official data before COVID-19 pandemic, show in healthy people with similar age of IPF patients, that a 70-year-old male today has about a third chance of dying by age 80, a 50% chance by age 84, and a 20% chance of reaching 90. A woman 70 years old has more than 80% chance of making it to age 80, 50% of making it to 87 and nearly 20% of making it to 93 [29].”.

Reviewer 3 Report

Dear Colleagues,

Thank you for the opportunity to get acquainted with this interesting work. My comments summarized below:

1.Need to carefully describe Statistical analysis (e.g. specify statistical criteria, p -value )

2. For all parameters, apart from mean and SD, the median and intraquartile range must be converted, as you are using a relatively small population

3.  For figures 1 and 2, you must specify the number of patients for each of the points. It is highly desirable to check the presence of stat. significant differences between median values using appropriate tests (for example, intraquartile regression or median regression)

4. It is highly desirable to perform a sensitivity analysis based on the male subpopulation (or population of smokers)

Author Response

Thank you for your letter and for the reviewer's comments on our manuscript titled “Nintedanib in Idiopathic Pulmonary Fibrosis: Tolerability and Safety in a Real-Life Single Center Experience in Patients Also Treating Oral Anticoagulant Therapy.” These comments are all valuable and very helpful in the review and improvement of our article as well as important guiding significance for our research. We have carefully reviewed the comments and made a correction, which we hope will meet your approval. The main corrections in the document and the answers to the reviewer's questions and comments are as follows.

Dear Colleagues,

Thank you for the opportunity to get acquainted with this interesting work. My comments summarized below:

1.Need to carefully describe Statistical analysis (e.g. specify statistical criteria, p -value )

R: The Statistical analysis section has been modified to better clarify the choice of the tests that were used. We also specified the statistical significance level. In agreement with reviewer’s comment, the paragraph was ameliorated as follows:

2.5 Statistical analysis. Continuous variables were summarized as mean ± standard deviation, while categorical variables were reported as percentages. The normality of continuous variables was tested with the Shapiro-Wilk test. As for spirometry measurements, values were compared at different time points using the paired t-test or the nonparametric Wilcoxon signed-rank test, when assumptions for the parametric test were not satisfied. When comparing measurements at baseline with those at 6 months, we considered only those individuals who had been taking NTDs for at least 4 months. When comparing measurements at baseline and at 2 months, we considered only individuals who had been taking NTD for at least 9 months. The comparison was carried in the restricted sample of male subjects. For FVC, we considered a non-responder subject after 6 months of NTD if FVC decreased > 5% and considered a differences were observed during the follow-up regarding blood test evaluations. Concomitant non-responder subject after 10 months of NTD if FVC decreased > 10%. Similarly, for DLCO we considered a non-responder subject after 6 months of NTD if DLCO decreased > 6% and we considered a non-responder subject after 10 months of NTD if DLCO decreased > 10% We performed a survival analysis considering the onset of NTD as a baseline and after 1 year of observation. The endpoint was death from any cause. Survival function was estimated by the Kaplan-Meier method. In all the analysis, the statistical significance was set at .05. Analyses were performed using R software version 4.2.1.

  1. For all parameters, apart from mean and SD, the median and intraquartile range must be converted, as you are using a relatively small population

R: In agreement with the reviewer’s observation, we included new table (Table 2) were spirometric measurements were summarized in terms of median and interquartile range.

Baseline

[median (IQR)]

6 months

[median (IQR)]

n

p-value

Baseline

[median (IQR)]

12 months

[median (IQR)]

n

p-value

FVC

76.0 (22.0)

76.0 (22.0)

31

0.6

76.0 (14.2)

76.0 (23.2)

22

0.2

‍FEV1

‍77.0 (23.0)

81.0 (24.0)

29

>0.9

82.5 (21.0)

74.0 (16.5)

20

0.039

‍DLCO

‍45.0 (28.0)

37.0 (20.0)

25

0.012

47.0 (30.8)

43.0 (30.2)

20

0.018

Table 2. Comparison of spirometric measurements after 6 months and after 12 months. 

  1. For figures 1 and 2, you must specify the number of patients for each of the points. It is highly desirable to check the presence of stat. significant differences between median values using appropriate tests (for example, intraquartile regression or median regression)

R: Thank you very much for this comment, the number of patients is now included in the figures. As for the statistical comparison, we applied a univariate statistical test for repeated measures. The suggestion of the Reviewer is very interesting, however our sample size is not large enough to perform a multivariate regression analysis (for example, the number of patients for 0-12 months FEV1 measurements is 20). We have included this point in the discussion as follows: “In the future, the increase in the numbers of patients enrolled, but above all the availability of PTF data, in particular in relation to the variation of FVC over time, it would possible to analyze the trend of lung function together with main clinical factors (eg. using a multivariate model) and this could surely help to define more precisely the effective therapeutic efficacy of NTD”.

  1. It is highly desirable to perform a sensitivity analysis based on the male subpopulation (or population of smokers)

R: As suggested, we performed the analysis restricted to the male subpopulation. Both in the 6 month- and 12 month-change, we only observed a significant decrease for DLCO (p=0.018 and p=0.020, respectively).

In agreement with reviewer’s comment we ameliorate this paragraph in the manuscript: “The same evaluation was made for differences between measurements at baseline and 12 months, which for FVC were available for 22 patients, for FEV1 for 22 patients and for DLCO for 20 patients (Figure 2 and 3). In this case, a statistically significant difference was detected for both FEV1 and DLCO (p-value respectively 0.039 and 0.018, Figure 2), while difference in FVC between baseline and 12 months expressed no statistical significance (p-value 0.2). The analysis restricted to male individuals gave similar results. Both in the 6 month- and 12 month-change, we only observed a significant decrease for DLCO (p=0.018 and p=0.020, respectively). For FVC, 6 (19%) were identified as non-responders at 6 months and 3 (14%) as non-responders at 12 months (Table 3). No association was found with smoking status, steroid, anti-antibody, PPI. For DLCO 13 (52%) were identified as non-responders at 6 months and 5 (25%) at 12 months. Equal in this case no association was found with smoking status, steroid, anti-antibody, PPI”.

Reviewer 4 Report

In this manuscript, Ruaro et. al. have examined the safety profile and efficacy of nintedanib in patients with IPF, including patients treated with anticoagulants. This is a retrospective evaluation of clinical data of IPF patients at a single center. Their study confirmed the efficacy and safety of nintedanib.

 While it is helpful to get further confirmation of the efficacy and safety profile of nintedanib in IPF, the major concern is novelty of this data. Larger clinical trials have well-established the efficacy and safety profile of nintedanib previously. Can they comment on what makes their study stand out in comparison to multiple other studies with similar results? Another limitation of this study is the small sample size.

Author Response

Comments reviewer’s: In this manuscript, Ruaro et. al. have examined the safety profile and efficacy of nintedanib in patients with IPF, including patients treated with anticoagulants. This is a retrospective evaluation of clinical data of IPF patients at a single center. Their study confirmed the efficacy and safety of nintedanib.

While it is helpful to get further confirmation of the efficacy and safety profile of nintedanib in IPF, the major concern is novelty of this data. Larger clinical trials have well established the efficacy and safety profile of nintedanib previously. Can they comment on what makes their study stand out in comparison to multiple other studies with similar results? Another limitation of this study is the small sample size.

Thank you for your letter and for the reviewer’s comments concerning our manuscript entitled "Nintedanib in Idiopathic Pulmonary Fibrosis: Tolerability and Safety in a Real Life Experience in a Single Centre in patients also treated with oral anticoagulant therapy”. Those comments are all valuable and very helpful for revising and improving our paper, as well as the important guiding significance to our research. We have reviewed the comments carefully and have made a correction, which we hope meets with approval. The main corrections in the paper and the responses to the reviewer’s comments are as follows.

Our study, as correctly highlighted by the reviewer, confirms the efficacy and safety of nintedanib in the treatment of idiopathic pulmonary fibrosis. The novelty of this study is the broadest possible view that we have given to the evaluation and follow-up of this disease. We have combined data related to a functional evaluation, obtained by examining tests of pulmonary function, with those of safety, not only by reviewing the actual known causes of discontinuation of nintedanib therapy, but also by associating its intake with that of therapies often used in subjects with multiple comorbidities, such as anticoagulants. Unlike most of the available studies on idiopathic pulmonary fibrosis, which have preferred to focus on one of these two aspects, we have tried to provide this different approach. The observation relating to the limited sample of patients is certainly correct. In agreement with this observation, we underline in the introduction that idiopathic pulmonary fibrosis is a rare disease characterized by disparate clinical courses with different speeds of progression; therefore, not all patients need antifibrotic therapies. In addition, we underline in the section limits this comment on sample size. In agreement with reviewer’s comments, we improve the following paragraphs in the manuscript: 1. INTRODUCTION. Idiopathic pulmonary fibrosis (IPF) is a rare and nonreversible chronic interstitial pneumonia characterized by abnormal deposition of extracellular matrix in the lung parenchyma [1-3]. Aberrant lung repair leads to repetitive tissue scar formation, alveolar structure abnormalities with significant impairment of alveolar gas exchange, and reduced lung function [forced vital capacity (FVC) and carbon monoxide diffusing capacity (DLCO)] [2,3]. IPF is the most common of its class (approximately 50-60% of all idiopathic interstitial pneumonia, with an estimated incidence of 3-9 cases per 100,000 individuals in Europe and North America [1-3]. It causes progressive exertional dyspnea and nonproductive cough, with rapid deterioration in the quality of life of patients with IPF. Although IPF exhibits a varied nature [4], clinic pathologic and histologic patterns allow the description of so-called habitual interstitial pneumonia (UIP) [5], defined radiologically on high-resolution computed tomography (HRCT) by a bilateral peripheral distribution of fibrosis and most pronounced at the bases of the lungs. The pathogenesis of these patterns is not entirely clear”; “4.DISCUSSION. The main limitation of our study is clearly the small sample size, as well as the even smaller number of repeated measurements. Despite this, it was possible to obtain evidence on the efficacy of NTD in terms of progression of the disease from a pulmonary function point of view as expected. Finally, it was not possible to assess whether stabilization of lung function was associated with better quality of life because of the absence of a systematic evaluation of this parameter. In the future, the increase in the numbers of patients enrolled, but above all the availability of PTF data, in particular in relation to the variation of FVC over time, could surely help to define more precisely the effective therapeutic efficacy of NTD”.

Round 2

Reviewer 2 Report

No more comments.

Reviewer 4 Report

Thank you for the revisions.